# OpenReview forum: "Reprogramming LLM Semantics: A Symbolic Attack Using Emoji-Based Context Mutation"
_ICLR.cc/2026/Conference — Submitted to ICLR 2026_

### Official Review · Reviewer_QHZ1 · 2025-10-21

**Soundness:** 1
**Presentation:** 1
**Contribution:** 1
**Rating:** 0
**Confidence:** 5

**Summary:**

This paper describes a text-to-text  _jailbreaking_ technique that uses an attacker LLM to transform a user prompt intended to elicit a harmful response so that it is less likely to be refused or deflected. Specifically, the technique instructs the attacker LLM to recast an instruction to the target LLM as an instruction to role-play as a player in a word game, substituting certain terms (nouns, verbs, URLs) by colon-enclosed emoji shortcodes. When the transformed prompts are not refused, the responses produced are not directly harmful but could be post-processed to attempt to reverse the transformation and recover an intelligible harmful response to the original prompt.

The authors evaluate the proposed _Emoji Game_ jailbreak technique on 100 prompts from the HarmBench dataset and compare the results against various black-box jailbreak techniques from the recent literature, measuring $\text{ASR}\_\text{ref}$, the proportion of non-refusal responses, judged by the absence of certain keywords, and $\text{ASR}\_\text{harm}$, the proportion of responses classified as harmful by an LLM-as-judge (gemma-3-12b-it).

**Strengths:**

- The evaluation setup includes a good coverage of models: reasoning, non-reasoning, open-weights, proprietary, and models from different providers.
- The evaluation of the refusal rate in responses (based on the occurrence of certain keywords) shows that responses to prompts using the proposed strategy are refused less often.
- The evaluation results using an LLM-as-judge harmfulness classifier are marginally better on one of the models under evaluation.

**Weaknesses:**

This paper proposes _Emoji Game_, a jailbreak technique that incorporates many ingredients from existing published or folklore attack strategies such as role-playing and encodings. When evaluated using an LLM-as-judge to measure the harmfulness of responses, it generally underperforms existing attacks (see Table 2) except on one of the models evaluated, gpt-oss-20b. As such, the contributions of the paper are iterative, limited to another variant of well documented and widely discussed jailbreak strategies, without advancing the understanding of the mechanisms why such strategies work, and without proposing any robust defense against them.

I encourage the authors to read [Rando2025], which eloquently articulates the characteristics that an academic paper on jailbreak techniques should possess to genuinely contribute towards the community understanding of safety vulnerabilities in LLMs and their mitigations.

[Rando2025] Javier Rando _Do not write that jailbreak paper_. In the Fourth Blogpost Track at ICLR 2025. Online: https://openreview.net/forum?id=TbN25IjHyC.

- Evaluation done exclusively on single-turn scenarios.
- Evaluation only includes black-box attack strategies as baselines, even on open-weight models or for techniques that are known to transfer to proprietary models like GCG.
- Missing key details to allow reproducibility (e.g. the system prompt of the harmfulness classifier).
- The paper often uses grandiose language without properly explaining its meaning (e.g. "[O]ur Emoji Game defines a precise symbolic grammar and a stable abstraction mechanism, allowing systematic reprogramming of the model interpretive layer.", "[W]e redefine the model’s understanding of meaning itself.", "[T]his is the first work to introduce a jailbreak framework based on formally structured symbolic grammar constructed through recursive emoji abstraction.")

**Questions:**

- What system prompt do you use to judge harmful responses using gemma-3-12b-it to report $\text{ASR}_\text{harm}$? Since you acknowledge on page 8 that "evaluating harm in this context requires a dedicated judge model that is specifically trained to recognize symbolic patterns and semantic camouflage used in the attack.", how can you be confident that gemma-3-12b-it is accurately judging the harmfulness of the responses your attack produces?

- On page 6, you say "Each strategy is executed at least 100 times per target model, though some run up to 300 trials depending on attack variability". Without any more details, my understanding is that you try at least each attack strategy (each baseline and yours) at least once on each of the 100 HarmBench prompts that you selected, but that sometimes you try each attack more than once based on the sample *variance* you observe (up to 3 times per prompt?). If this is correct, you could be sometimes comparing pass@3 with pass@1 results. Can you clarify what you meant and why you did not just run each attack on each prompt consistently 3 times reporting the variance?

- What system prompt do you use to post-process the responses of the target model using GPT-5? Did you try to evaluate the harmfulness of post-processed responses using the same method you use to evaluate the original responses?

---

### Official Review · Reviewer_YcTZ · 2025-10-27

**Soundness:** 2
**Presentation:** 1
**Contribution:** 2
**Rating:** 2
**Confidence:** 4

**Summary:**

The paper introduces a jailbreaking method that encodes a malicious prompt as compositions of symbolic primitives in a fictional “Emoji” language, while explicitly instructing the LLM that these primitives are non-executable. The target model is then asked to produce its (malicious) response in this same language. An attacker LLM defines the language at runtime for each query, using three primitive types. On evaluation, the method generally outperforms baseline attacks on refusal-based attack success rate (ASR) metrics and is competitive on a HarmBench-derived ASR metric.

**Strengths:**

The paper is original in proposing a novel obfuscation strategy to encode malicious queries as primitives in a fictional game language. Based on the provided examples, the resulting model outputs are detailed and, when mapped back to natural language, unambiguously harmful - showing the attack’s practical potency. Empirically, the method generally outperforms or remains competitive with the baselines reported in the paper. The work is timely given increasing misuse of LLMs (e.g., misinformation).

**Weaknesses:**

- The approach presented in the paper is conceptually not very novel - similar methods have been proposed in papers such as [1][2][3]. Emoji Game can be construed as another instantiation of a Cipher attack, and does not use these other highly-related attacks as baselines.
- The manuscript itself appears heavily AI-written, leading to degraded clarity - non-standard terminology, formatting issues (e.g., the citation on line 85), and redundancy (the last two paragraphs of Related Work convey essentially the same content).
- While the paper contrasts itself with prior work in terms of methodology, it does not convincingly motivate why Emoji Game is needed as a distinct approach.
- The work is outperformed by existing methods when evaluated using the more standard HarmBench-based metric. It is only more performant than existing methods when using the refusal-based ASR metric, which is a weaker metric in my opinion. The paper also does not mention whether this metric aligns with human safety preferences.
- The paper does not benchmark itself against any standard defense procedures. In particular, since the output can be interpreted by an LLM to retrieve harmful content, I suspect that output filtering will prove to be quite effective against Emoji Game.
- I feel that there is a lot of information missing in the paper, which I will enumerate in the “Questions” section below.

[1] GPT-4 Is Too Smart To Be Safe: Stealthy Chat with LLMs via Cipher. Youliang Yuan, Wenxiang Jiao, Wenxuan Wang, Jen-tse Huang, Pinjia He, Shuming Shi, Zhaopeng Tu.
[2] ArtPrompt: ASCII Art-based Jailbreak Attacks against Aligned LLMs. Fengqing Jiang, Zhangchen Xu, Luyao Niu, Zhen Xiang, Bhaskar Ramasubramanian, Bo Li, Radha Poovendran.
[3] Making Them Ask and Answer: Jailbreaking Large Language Models in Few Queries via Disguise and Reconstruction. Tong Liu, Yingjie Zhang, Zhe Zhao, Yinpeng Dong, Guozhu Meng, Kai Chen.

**Questions:**

- What is motivation for the specific refusal words chosen by the authors for the refusal-based ASR? Are they taken from prior work?
- Are the primitives for the Emoji Game regenerated for each query? What is the token cost per attack?
- What does ‘symbolic plausibility’ mean in line 255?
- How are the ‘interpretations’ generated in line 263?
- ‘Appendix 5’ isn’t in the paper in line 340.
- The paragraph starting at line 322 is unclear - what is ‘trial size’?
- The terminology used in the tables is not properly defined and explained.
- The paper mentions improvements in multi-turn settings at multiple places - are there any experiments backing this claim?

---

### Official Review · Reviewer_fhPM · 2025-10-30

**Soundness:** 2
**Presentation:** 1
**Contribution:** 1
**Rating:** 0
**Confidence:** 5

**Summary:**

This work studies jailbreak attacks that encode malicious instructions as emoji. The attacker converts prompts into an emoji language, performs Q&A with the model in that emoji environment, and then decodes the model’s emoji outputs back into harmful content. While the idea is interesting and the authors report high attack success rates, the manuscript reads more like an experimental report than a conference paper and, in my opinion, does not meet ICLR standards for acceptance.

**Strengths:**

The idea is interesting and uncovers a new safety risk: emojis can be used as a covert channel to convey malicious instructions. The authors demonstrate the feasibility of this approach and report reasonably strong attack success rates.

**Weaknesses:**

1. The writing is verbose and unfocused. The background, core idea, and experiments could be presented concisely — I estimate the content could be reduced to 4–6 pages rather than the current 9.

2. Limited novelty. Conceptually, this work differs from prior work mainly in the choice of a different proxy “language” (emoji). Prior papers have used alternate encodings; switching to emojis feels like an incremental variation rather than a substantive advance.

3. A major technical concern: I remain unconvinced that arbitrary text can reliably be encoded as emojis, or that all models handle emojis well. If either assumption fails — that is, if some texts cannot be faithfully represented as emoji sequences, or if target models poorly understand emoji — then the proposed attack would break. The paper does not sufficiently address this fragility across different models and inputs.

**Questions:**

Refer to the proposed weakness.

---

### Meta-Review · Area_Chair_BxMr · 2026-01-06

**Summary:**

This paper shows that emoji-based symbolic encoding can bypass refusals in LLMs.

Reviewer consistently raise serious concerns about limited novelty relative to prior obfuscation/cipher-style jailbreaks, poor presentation and clarity, overstated claims about "semantic reprogramming", and methodological weaknesses, including reliance on a weak refusal-based ASR metric, insufficient comparison to closely related baselines, missing experimental details and lack of analysis of robustness, defenses and reproducibility.

I cannot accept the paper in the current form.

**Reviewer Concerns:**

No rebuttal from authors.

**Reviewer Scores:**

Many keys issues raised by reviewers. No discussion can revert the ratings.

---

### Decision · Program_Chairs · 2026-01-26

Reject